# Cholesterol-Lowering Drugs as Potential Antivirals: A Repurposing Approach against Flavivirus Infections

**DOI:** 10.3390/v15071465

**Published:** 2023-06-28

**Authors:** Juan Fidel Osuna-Ramos, Carlos Noe Farfan-Morales, Carlos Daniel Cordero-Rivera, Luis Adrián De Jesús-González, José Manuel Reyes-Ruiz, Arianna M. Hurtado-Monzón, Selvin Noé Palacios-Rápalo, Ricardo Jiménez-Camacho, Marco Antonio Meraz-Ríos, Rosa María Del Ángel

**Affiliations:** 1Department of Infectomics and Molecular Pathogenesis, Center for Research and Advanced Studies (CINVESTAV-IPN), Mexico City 07360, Mexico; carlos.farfan@cinvestav.mx (C.N.F.-M.); carlos.cordero@cinvestav.mx (C.D.C.-R.); luis.dejesus@cinvestav.mx (L.A.D.J.-G.); arianna.hurtado@cinvestav.mx (A.M.H.-M.); selvin.palacios@cinvestav.mx (S.N.P.-R.); ricardo.jimenez@cinvestav.mx (R.J.-C.); 2Facultad de Medicina, Universidad Autónoma de Sinaloa, Culiacán 80019, Mexico; 3Departamento de Ciencias Naturales, Universidad Autónoma Metropolitana (UAM), Unidad Cuajimalpa, Mexico City 05348, Mexico; 4Unidad de Investigación Biomédica de Zacatecas, Instituto Mexicano del Seguro Social, Zacatecas 98000, Mexico; 5Unidad Médica de Alta Especialidad, Hospital de Especialidades No. 14, Centro Médico Nacional “Adolfo Ruiz Cortines”, Instituto Mexicano del Seguro Social (IMSS), Veracruz Norte, Veracruz 91810, Mexico; jose.reyesr@imss.gob.mx; 6Facultad de Medicina, Región Veracruz, Universidad Veracruzana (UV), Veracruz 91090, Mexico; 7Departamento de Biomedicina Molecular, Centro de Investigación y de Estudios Avanzados del Instituto Politécnico Nacional (CINVESTAV-IPN), Mexico City 07360, Mexico; mmeraz@cinvestav.mx

**Keywords:** flaviviruses, DENV, ZIKV, YFV, cholesterol-lowering drugs, antiviral agents, atorvastatin, ezetimibe, synergistic effect, therapeutic use

## Abstract

Flaviviruses, including Dengue (DENV), Zika (ZIKV), and Yellow Fever (YFV) viruses, represent a significant global health burden. The development of effective antiviral therapies against these viruses is crucial to mitigate their impact. This study investigated the antiviral potential of the cholesterol-lowering drugs atorvastatin and ezetimibe in monotherapy and combination against DENV, ZIKV, and YFV. In vitro results demonstrated a dose-dependent reduction in the percentage of infected cells for both drugs. The combination of atorvastatin and ezetimibe showed a synergistic effect against DENV 2, an additive effect against DENV 4 and ZIKV, and an antagonistic effect against YFV. In AG129 mice infected with DENV 2, monotherapy with atorvastatin or ezetimibe significantly reduced clinical signs and increased survival. However, the combination of both drugs did not significantly affect survival. This study provides valuable insights into the potential of atorvastatin and ezetimibe as antiviral agents against flaviviruses and highlights the need for further investigations into their combined therapeutic effects.

## 1. Introduction

Arboviruses, particularly members of the Flavivirus genus, such as the Dengue (DENV), Zika (ZIKV), and Yellow Fever (YFV) viruses, are well-known for their extensive geographical distribution and significant morbidity and mortality effects [1,2]. These viruses are primarily transmitted through the bite of infected *Aedes* mosquitos, resulting in a wide range of clinical manifestations ranging from mild febrile illnesses to severe and potentially fatal conditions such as Dengue Hemorrhagic Fever/Dengue Shock Syndrome, Zika-induced microcephaly, and Yellow Fever-induced hepatic failure [1,2,3,4,5].

In flavivirus infection, the role of cholesterol, an essential structural component of lipid rafts and host cell membranes, has received considerable attention in recent years [6,7]. Cholesterol has been linked to several stages of flavivirus infection, including viral entry, replication, and egress [6]. For this reason, flaviviruses hijack the host’s cholesterol machinery to support their life cycle and replication [8,9,10]. In this context, modulating cholesterol homeostasis in host cells could present a potential therapeutic target for flavivirus infections [7,8,11,12].

Notably, drug repurposing, which involves identifying new therapeutic uses for already-approved drugs, is a promising approach in antiviral therapy, potentially reducing the time and cost of conventional drug development, which can also be used in emerging viral infections [13,14,15,16]. Atorvastatin and ezetimibe, two well-established and FDA-approved cholesterol-lowering drugs, have shown promising antiviral properties against various viruses, including flaviviruses [7,10,17], Hepatitis B virus (HBV) [18] and Human Immunodeficiency Virus-1 (HIV-1) [19]. These drugs disrupt the formation or absorption of cholesterol, thereby perturbing the lipid rafts and replicative complex formation [6,7].

Previous in vitro studies have shown that atorvastatin and other statins, such as HMG-CoA reductase inhibitors, can reduce DENV and ZIKV replication [8,11,17,20]. Similarly, ezetimibe, which inhibits cholesterol absorption by blocking the Niemann-Pick C1-like 1 (NPC1L1) protein, has been shown to impair the replication of Hepatitis C and DENV by limiting the cholesterol necessary for viral replication [10,21]. This evidence suggests that cholesterol-lowering drugs could be used as a new class of antivirals against flaviviruses; however, additional research is necessary to fully elucidate their therapeutic potential and underlying mechanisms. The objective of the current study was to assess the efficacy of atorvastatin and ezetimibe alone or in combination against DENV, ZIKV, and YFV in an in vitro model and against DENV in an in vivo model.

## 2. Materials and Methods

### 2.1. Cell Culture, Virus, and Reagents

The human hepatoma-derived cell line Huh-7 (kindly donated by Dr. Ana Maria Rivas, Autonomous University of Nuevo León) was grown in advanced DMEM (complete medium) supplemented with 2 mM glutamine, penicillin (5 × 10^4^), streptomycin (50 µg/mL), 8% fetal calf serum (FCS), and 1 mL/L of amphotericin B (Fungizone) at 37 °C and a 5% CO_2_ atmosphere.

The propagation of DENV serotype 2 (New Guinea strain) and DENV serotype 4 (H241 strain) was carried out using CD-1 suckling mice brains (provided by the Laboratory Animal Production and Experimentation Unit at Cinvestav (UPEAL for its Spanish acronym)). ZIKV (strain MEX_CIENI551, provided by Dr. Jesús Torres, Escuela Nacional de Ciencias Biológicas, Instituto Politécnico Nacional, Mexico) and YFV (vaccine strain 17D) were propagated in C6/36 mosquito cells. The Brain extracts of MOCK-infected CD-1 suckling mice were used as a control, as previously described [22].

In flow cytometry and confocal microscopy, anti-prM-E monoclonal 2H2 (ATCC^®^ HB-114) and anti-Flavivirus group antigen 4G2 (ATCC^®^ VR-1852) antibodies, anti-E (Genetex, Irvine, CA, USA), anti-NS3 (Genetex), and anti-C (Genetex) antibodies were used as reagents to stain DENV viral proteins. The concentrations of the antibodies were determined based on the manufacturer’s instructions and previous titration experiments. Ezetimibe (Cayman Chemical, Ann Arbor, MI, USA) and atorvastatin (Sigma-Aldrich, St. Louis, MO, USA) were utilized for the inhibition assays.

### 2.2. Flavivirus Infection and Treatment

The Huh-7 cells were seeded at 70–80% confluence in 24-well plates and infected with DENV (serotype 2 or 4), ZIKV, or YFV at a multiplicity of infection (MOI) of 3 in medium supplemented with 1% FCS for 2 h at 37 °C. The cells were then washed three times with Hanks’ solution, and an inhibition assay was performed using ethanol as a vehicle or ezetimibe in complete medium for 48 h at 37 °C Untreated infected cells and non-infected cells served as controls in the assessment. The inhibition assay was performed in triplicate, and the mean values with standard error of the mean (SEM) were reported.

### 2.3. Flow Cytometry

Flow cytometry was used to determine the proportion of Huh-7-infected untreated and treated cells that were infected. The harvested cells were fixed with 1% formaldehyde, permeabilized with a permeabilizing solution (0.1 percent saponin, 1 percent SFB, and 1× PBS) for 20 min, and they were incubated with anti-prM-E (2H2 or 4G2) antibodies for 2 h at room temperature (RT). Goat anti-mouse Alexa Flour 488 (Life Technologies, Carlsbad, CA, USA) and goat anti-mouse Alexa Flour 405 (Life Technologies) antibodies were utilized as secondary antibodies. Flow cytometry was performed on a BD LSR FortessaTM cytometer, and the FlowJo v. 10 software was used to analyze the data.

### 2.4. Cell Viability Assay, CC_50_, IC_50_, and Selectivity Index (SI) Determination

Using propidium iodide (PI) and MTT, the cell viability of cells treated with vehicle or increasing concentrations of ezetimibe (0, 5, 10, 15, 25, 35, 45, 50, and 60 µM) or atorvastatin (0, 5, 10, 15, 25, 35, 45, 50, and 60 µM) alone or in combination (5, 10, and 20 µM for atorvastatin and 13, 20, and 50 µM for ezetimibe) for 48 h at 37 °C was determined. The MTT method was evaluated by spectrophotometry (BioTek ELx800TM, Winooski, VT, USA) measuring the absorbance at 540 nm, while the PI method was evaluated by flow cytometry as the percentage of live cells. Infected cells were treated with vehicle or increasing concentrations of atorvastatin or ezetimibe against DENV 2, DENV 4, ZIKV, and YFV. The 50% cytotoxic concentration (CC_50_) and inhibitory concentration (IC_50_) were calculated. Both CC_50_ and IC_50_ are calculated by plotting x-y and fitting the data to a straight line (linear regression) after logarithmic transformation of the concentration and normalization defining 0% as the smallest mean in each data set and 100% as the largest mean in each data set. The selectivity index (SI) was calculated using the CC_50_/IC_50_ ratio.

### 2.5. Drug Combination Assay and Synergism

Huh-7 cells were infected with DENV, ZIKV, or YFV at a multiplicity of infection (MOI) of 3 and treated at the following concentrations: 5, 13, and 10 µM for atorvastatin and 13, 20, and 50 µM for ezetimibe. A DMSO–ethanol mixture was used as control. Following 48 h of treatment, cell viability was evaluated using the MTT assay, as indicated previously. After that, synergism tests were conducted. To calculate synergism, a matrix of sixteen different combinations of atorvastatin and ezetimibe concentrations were used, and the experiment was performed in duplicate using a dose–response matrix in 24-well plates. To determine whether drug combinations act synergistically against flavivirus infection, the observed responses were compared to the expected responses of the combination. The expected responses were calculated using the R package SynergyFinder based on Loewe’s additive reference model [23]. RStudio version 4.1.0 of R programming language was used to visualize the synergism results as response–dose matrices and distribution heatmaps to represent the synergism score.

### 2.6. Treatment and Survival Assays in the AG129 Mouse Model

In the AG129 mouse model infected with DENV 2 and treated with ezetimibe or atorvastatin alone or in combination, infection and treatment assays were conducted. Treatments were initiated on day four post virus inoculation (p.i.) with administration of the drug every 24 h for ezetimibe (10 mg/kg/d) and every 12 h for atorvastatin (20 mg/kg/d) as monotherapy or in combination (ezetimibe + atorvastatin: 10 + 20 mg/kg/d) for six days (day 10 p.i.). Before being administered by gastric cannula, drugs were diluted in physiological solution and sprayed. We worked exclusively with 6- to 8-week-old, 23- to 25-g male mice. Monitoring was conducted daily for surveillance and evaluation of morbidity indicators, according to the Orozco et al. morbidity scale [24], the percentage of weight change and clinical signs were also evaluated daily. When the animals reached a score of 5, they were euthanized in a CO_2_ chamber. Mice that died of causes other than infection were omitted from the experiment.

### 2.7. Statistics Analysis

Continuous variables were represented using the mean and the standard error of the mean (SEM). To determine differences between concentrations used in inhibition assays with ezetimibe and atorvastatin in monotherapy or in combination, the nonparametric Kruskal–Wallis test was adjusted for multiple comparisons with Dunn’s post hoc test. The IC_50_ and CC_50_ values were calculated using nonlinear regression. Statistical analysis was performed with Graph Pad Prism version 6 software. Using the additive reference model of Loewe [25] and the R package SynergyFinder [23], the potential synergistic effect was evaluated. Multiple mean comparisons were conducted using the Wilcox test to compare each treatment group to the control in the in vivo model. Survival was analyzed with Kaplan–Meier curves and compared with log-rank tests using R version 4.1.0 and Rstudio version 1.3, respectively. At the 95% confidence level (*p* < 0.05), statistical significance was determined.

### 2.8. Ethical Statements

This study was conducted according to the Official Mexican Standard Guidelines for Production, Care, and Use of Laboratory Animals (NOM-062-ZOO-1999), and the protocol number 048–02 was approved by the Animal Care and Use Committee (CICUAL) at CINVESTAV-IPN, Mexico.

## 3. Results

### 3.1. Effect of Cholesterol-Lowering Drugs Atorvastatin and Ezetimibe in Monotherapy on In Vitro DENV, ZIKV, and YFV Infection

First, the viability of cells during treatment with various concentrations of ezetimibe and atorvastatin was assessed. During analysis, we found that ezetimibe and atorvastatin had no effect on cell viability as measured by IP and MTT methods (Figure 1A,B), and CC_50_ values were calculated for both drugs (Figure 1C,D). After assessing cellular viability, we found that ezetimibe (Figure 2A,C) and atorvastatin (Figure 2B,D) were able to reduce the percentage of infected cells with DENV 2, DENV 4, ZIKV, or YFV in a dose-dependent manner.

The IC_50_ for ezetimibe was obtained at 48 h post-infection based on a previously published study, yielding values of 19.15 µM (95% CI = 16.71–21.95 µM) and 17.68 µM (95% CI = 15.99–19.56 µM) for DENV 2 and 4, respectively [10]. Concerning ZIKV and YFV, the respective IC_50_ values were 24.16 µM (95% CI = 19.43–26.35 µM) and 23.65 µM (95% CI = 20.12–25.78 µM) (Figure 2A).

On the other hand, the atorvastatin inhibitory concentration at 50% was calculated, revealing that for DENV 2 and 4, the IC_50_ values were 24.12 µM (95% CI = 21.91 to 26.55 µM) and 12.19 µM (95% CI = 9.948 to 14.95 µM), respectively (Figure 2B). For ZIKV, the atorvastatin IC_50_ value was 25.13 µM (95% CI = 23.84 to 28.95 µM), and it was 17.34 µM (95% CI = 14.22 to 19.87 µM) for YFV (Figure 2B).

Once the IC_50_ was determined and with the CC_50_ values previously established with the viability assay, the selectivity index (SI) was calculated for each virus and both drugs, which are shown in Table 1.

### 3.2. Effect of Cholesterol-Lowering Drugs in Combination on In Vitro DENV, ZIKV, and YFV Infection

Viral infection induces an increase in cholesterol absorption and synthesis to support flavivirus replication [6,11]. For this reason, we considered that the combination of the drugs atorvastatin and ezetimibe, which separately inhibit cholesterol synthesis and absorption, may induce a synergic effect and result in more efficient HDA therapy for flavivirus inhibition. Three combinations of atorvastatin and ezetimibe that did not affect cell viability were selected (Figure 3A). The best anti-flaviviral effect was found in the combination of atorvastatin 20 µM and ezetimibe 50 µM (Figure 3C–F), suggesting that the combination of these drugs may have a potential therapeutic effect during infections by the flavivirus DENV, ZIKV, or YFV.

### 3.3. Combination and Synergism

Matrix-based experiments were designed using the following experimental strategy: in vitro infection with DENV, ZIKV, and YFV as well as treatment with a combination of cholesterol-lowering drugs. To test a potential synergy between atorvastatin and ezetimibe, the effect was analyzed with the programming and statistical tool R and the Synergy Finder package [23]. Based on the Loewe method, different synergism scores between ezetimibe and atorvastatin against various flaviviruses were determined in this study. Additionally, distribution maps were created based on these scores to represent the likelihood that drugs in combination will have a synergistic, additive, or antagonistic effect. The purpose of this determination is to identify a synergistic effect when the Loewe score exceeds 10, an additive effect when the score is between −10 and 10, and an antagonistic effect when the score is less than −10 [23].

For DENV 2, the combination of atorvastatin and ezetimibe demonstrated high inhibition percentages, as depicted in this distribution map region, resulting in a higher Loewe synergy score of 21.67. (Figure 4A). Additionally, for ZIKV, a Loewe synergism score of 7.04 was obtained, indicating a region that, according to distribution maps, exhibited the highest synergism along against DENV 2. (Figure 4C). On the other hand, for the other viruses, both drugs demonstrated a smaller region of synergism, yielding scores of −1.18 for DENV 4 and −12.17 for YFV (Figure 4B).

In summary, we found that the combination of atorvastatin and ezetimibe obtained a synergistic effect against DENV 2, while an additive effect was observed for DENV 4 and ZIKV and an antagonistic effect was observed for YFV.

### 3.4. The Effect of Monotherapy and Combination Therapy with Ezetimibe and Atorvastatin on DENV 2-Infected AG129 Mice

Finally, the effect of cholesterol-lowering drugs on flavivirus infection in AG129 mice was assessed. Based on the in vitro synergism test results, we decided to examine the antiviral effect against DENV 2. Infected and uninfected mice were treated with doses of ezetimibe 10 mg/kg/d and atorvastatin 20 mg/kg/d in monotherapy and in combination ezetimibe + atorvastatin: 10 + 20 mg/kg/d beginning on day 4 post-infection (p.i). To avoid interventions that could alter the course of the experiment, only clinical signs, weight loss, and survival were assessed (Figure 5A,B and Figure 6A,B). Daily weight measurements were performed on all mice to calculate the percentage of weight loss before infection and treatment as well as to conduct a Kaplan–Meier analysis for survival analysis (Figure 6A,B).

In the treatment trial involving AG129 mice, a total of 25 animals were examined: 10 control mice without treatment, 5 mice treated with ezetimibe, 5 mice treated with atorvastatin, and 5 mice treated with both ezetimibe and atorvastatin. The daily documentation of clinical signs revealed a significant decline in the frequency of clinical indicators (Figure 5A). On day 10 post-intervention (day 7 of therapy), both atorvastatin and ezetimibe had statistically significant effects (*p* < 0.05) (Figure 5A,B). Compared to the infected and untreated control group, only with atorvastatin was the weight maintained on day 10 pi, thereby preventing significant weight loss with atorvastatin (Figure 5B).

On the other hand, survival was evaluated without animal interventions that could have altered the outcome. Weight and comorbidity indicators were evaluated daily. The survival rate of infected mice treated with a single dose of atorvastatin and ezetimibe was significantly increased (*p* = 0.0016) (Figure 6B). According to the Kaplan–Meier survival analysis, the median survival times for mice treated with ezetimibe and atorvastatin were 13 and 14 days, respectively. The log-rank test revealed a statistically significant difference between the survival medians of ezetimibe (*p* = 0.005) and atorvastatin (*p* = 0.0002) when compared to the untreated control group (median survival: 11 days) (Figure 6B). To our surprise, the combination of the two drugs had no effect on the increased survival of DENV 2-infected mice (*p* = 0.16).

## 4. Discussion

Flavivirus infections are still a worldwide concern for public health, and yet effective treatments remain elusive [1,7,26]. A promising strategy is host-directed antiviral therapy (HDA), which targets host factors exploited by viruses for replication. This strategy is particularly compelling due to the genetic austerity of viruses, which requires their reliance on host cellular elements and organelles to complete their life cycle. A critical host factor, cholesterol, is a key component of cell membranes and plays a pivotal role in the flavivirus replicative cycle [6,11,12]. Consequently, this molecule has been proposed as a therapeutic target for flavivirus infections, with a focus on the repurposing of FDA-approved cholesterol-lowering drugs with known antiviral effects, such as atorvastatin and ezetimibe [7]. 

Statins, such as atorvastatin, inhibit the cholesterol biosynthetic pathway by functioning as structural analogs of HMG-CoA, an intermediate metabolite in the mevalonate pathway [27]. As such, they can interact with and competitively inhibit the HMGCR enzyme responsible for cholesterol synthesis [28]. This mechanism has particular relevance for flavivirus treatment strategies, as previous research has suggested that statins are being used to treat infections caused by hepatitis C virus (HCV), Japanese encephalitis virus (JEV), Influenza A virus (IAV), and severe acute respiratory syndrome coronavirus 2 (SARS-CoV-2) [29,30,31,32]. In fact, cholesterol depletion by statins has been shown to reduce the infectivity and viral production of IAV and respiratory syncytial virus (RSV) as well as the viral entry, fusion, and replication processes of human immunodeficiency virus type 1 (HIV-1) and HCV [32,33,34,35,36,37].

Interestingly, ezetimibe, another cholesterol-lowering drug, has been underexplored for its antiviral effect. It targets the cholesterol receptor NPC1L1 [38] and operates by blocking cholesterol-induced internalization of NPC1L1 [39]. To date, only one report, published by our research group, has documented the cholesterol-dependent anti-DENV effect of this drug, with no other studies reporting antiviral effects against other arboviruses [10]. However, evidence also points to ezetimibe’s efficacy in reducing HCV, HBV, and EBOV infection, although the antiviral effect appears to be related more to the ability of this drug to block viral entry receptors than to its hypolipidemic effect [18,21,40,41].

In the current investigation, the individual antiviral effects of atorvastatin and ezetimibe were assessed using an in vitro model in Huh-7 cells. The treatment with atorvastatin resulted in a significant and variable reduction in the percentage of infected cells in a dose-dependent manner for different flaviviruses, thus corroborating previous findings [17,42,43,44]. Similarly, ezetimibe demonstrated a broad anti-flavivirus effect, which aligns with prior research by our group and others that reported similar effects [10,18,21]. The synergistic effect was not observed for all flaviviruses examined. In addition to inhibiting statin-induced cholesterol absorption, the combination of atorvastatin and ezetimibe may have an additive anti-DENV effect. The combination of atorvastatin and ezetimibe against DENV 2 had only a synergistic effect. This result compels us to conduct in vivo tests to determine the antiviral efficacy of atorvastatin and ezetimibe against DENV 2. DENV 2 is one of the four dengue virus serotypes known to cause severe disease in humans [45]. By concentrating on DENV 2, we intended to investigate therapies for one of the virus’s most dangerous forms. However, it is plausible that similar methods could be utilized to study in vivo infection of additional flaviviruses. This may be due to flaviviruses’ use of diverse host factors, as well as variations in virus–host interactions, which can be influenced by different host cell types and virus strains. [6,46,47,48,49]. Therefore, even though the combined treatment shows promise, it may not be applicable to all flavivirus infections and requires further study.

To date, there are limited studies with combination treatments for flavivirus infections. What has been observed so far in the AG129 mouse model, which is permissive for DENV infection, is that lovastatin treatment resulted in a two-day delay in virus-induced mortality, independent of the time at which treatment was initiated [20]. As for ZIKV, in vitro assays have shown that different statins effectively inhibit ZIKV replication [17,42]. Interestingly, only lipophilic statins showed anti-ZIKV effects, suggesting lipophilicity is a crucial antiviral property [17]. In this regard, there is evidence that the lipophilicity of statins is related to the specificity, efficacy, and pleiotropic effects of these drugs because it allows interaction with lipid membranes [42]. However, the role of structure and biophysical properties in the antiviral effects of statins has been poorly studied, and no in vivo studies confirm that these antiviral properties of the drugs are limited to lipophilic statins.

In the present study, we found a reduction in the appearance of clinical signs in DENV 2-infected mice treated with both atorvastatin and ezetimibe monotherapy. This effect was observed from day 10 after inoculation and on the seventh day of treatment. Weight reduction was also significantly avoided in animals infected with DENV 2 and treated with atorvastatin. Regarding survival, a significant increase in survival was achieved in mice infected with DENV 2 and treated with single doses of atorvastatin and ezetimibe. No significant weight reduction, delay in the onset of clinical signs, or increase in survival was found using the combination of both drugs. The latter could be explained by compensatory mechanisms that may be occurring in the body since it has been observed that NPC1L1 expression, for example, is specific and exclusive to some tissues such as the liver and small intestine of mice [50,51]. Therefore, ezetimibe, through inhibition of NPC1L1, may cause a decrease in the ability of cells to uptake free cholesterol, increasing the activity of HMG-CoA reductase, the rate-limiting enzyme in cholesterol synthesis [52], which somewhat offsets the inhibitory effect that ezetimibe and atorvastatin in combination might have. Notably, atorvastatin in monotherapy had a significant antiviral effect in AG129 mice infected with DENV 2, as seen with other statins such as lovastatin. Atorvastatin, like lovastatin, is considered a lipophilic statin, but unlike lovastatin, atorvastatin has a higher bioavailability that can reach up to 12% and a half-life of up to 14 h [20]. It further enables shorter treatments and more effective doses, which may result in fewer side effects. The above characteristics make it ideal for treating acute infections such as those caused by DENV 2 infections.

## 5. Conclusions

The present research suggests that the cholesterol-lowering drugs atorvastatin and ezetimibe could potentially treat flavivirus infections. In vitro assays showed that atorvastatin and ezetimibe had a synergistic effect against DENV 2, additive effects against DENV 4 and ZIKV, and an antagonistic impact on YFV. An infection model in AG129 mice demonstrated a significant reduction in the onset of clinical signs and the maintenance of weight percentage. This preclinical evidence could pave the way for future randomized controlled clinical trials involving patients infected with these flaviviruses and could also provide a solid foundation for furthering the preclinical evaluation of the effectiveness of these drugs.

## Figures and Tables

**Figure 1 viruses-15-01465-f001:**
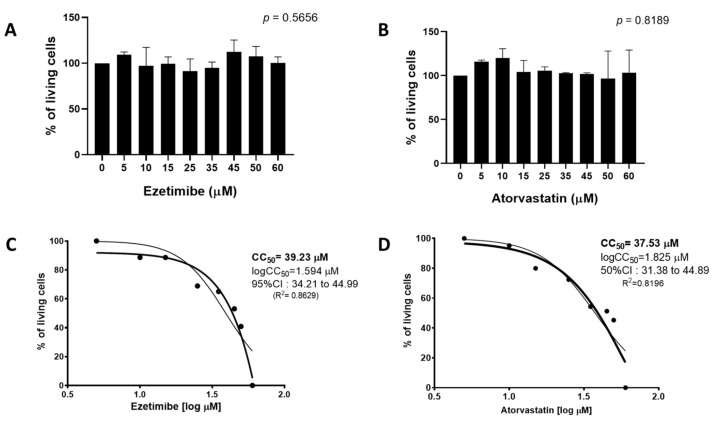
The effects of ezetimibe and atorvastatin on cell viability in monotherapy. The effect of ezetimibe (**A**) and atorvastatin (**B**) concentrations on the percentage of living Huh-7 cells after 48 h of treatment by IP. The dose–response curves representing the CC_50_ for ezetimibe (**C**) and atorvastatin (**D**), the CC_50_ was determined by an MTT assay. The error represents the standard error of the mean (SEM), and the results are displayed as bar graphs. The CC_50_ was represented as normalized logarithmic dose–response curves.

**Figure 2 viruses-15-01465-f002:**
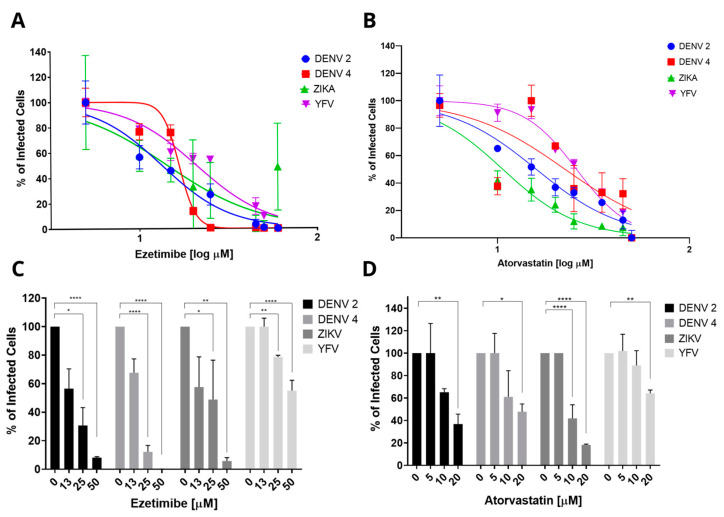
Antiviral effect of ezetimibe and atorvastatin. The dose–response curves representing the IC_50_ of ezetimibe (**A**) and atorvastatin (**B**) in monotherapy against DENV 2, DENV 4, ZIKV, and YFV (**A**). The percentage of HUH-7 infected cells with DENV 2, DENV 4, ZIKV, and YFV following treatment with 13, 25, and 50 μM of ezetimibe and vehicle (0 μM) (**C**) and 5, 10, and 20 μM of atorvastatin and vehicle (0 μM) (**D**). The asterisk values indicate the statistical significance of the value of * *p* = 0.05, ** *p* = 0.005, and **** *p* < 0.0001.

**Figure 3 viruses-15-01465-f003:**
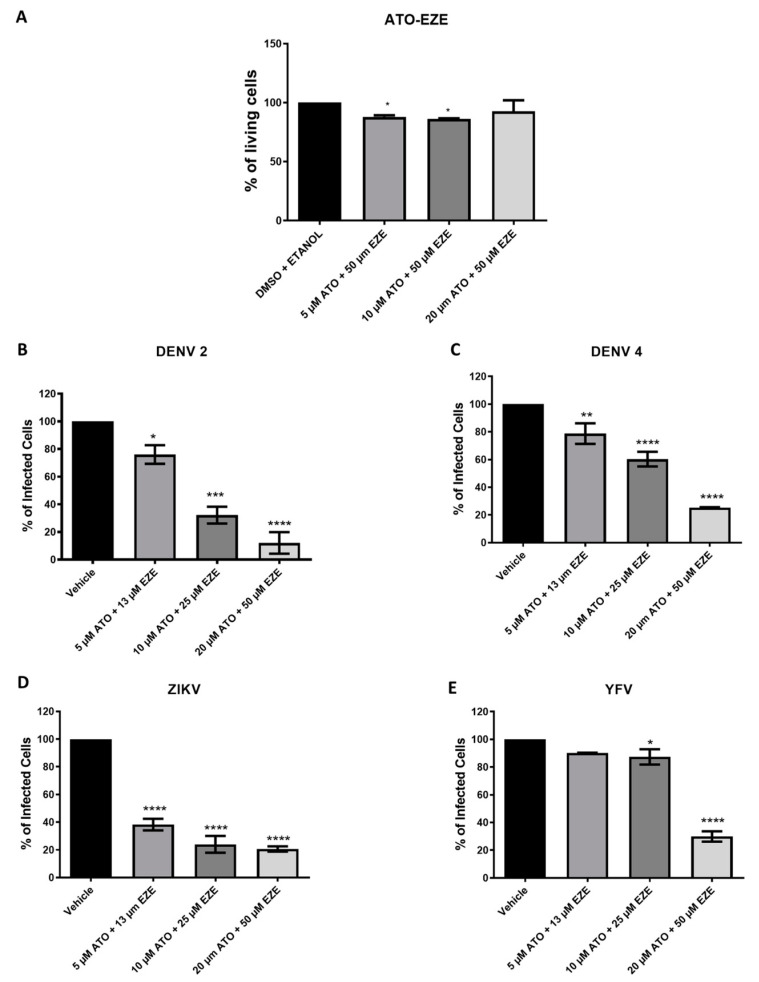
Cell viability of atorvastatin and ezetimibe drugs in combination (**A**). Bar plots show the antiviral effect of atorvastatin and ezetimibe in combination against (**B**) DENV 2, (**C**) DENV 4, (**D**) ZIKV, and (**E**) YFV. Asterisks values represent statistical significance of * *p* value = 0.05, ** *p* value = 0.01, *** *p* = 0.001, and **** *p* < 0.0001.

**Figure 4 viruses-15-01465-f004:**
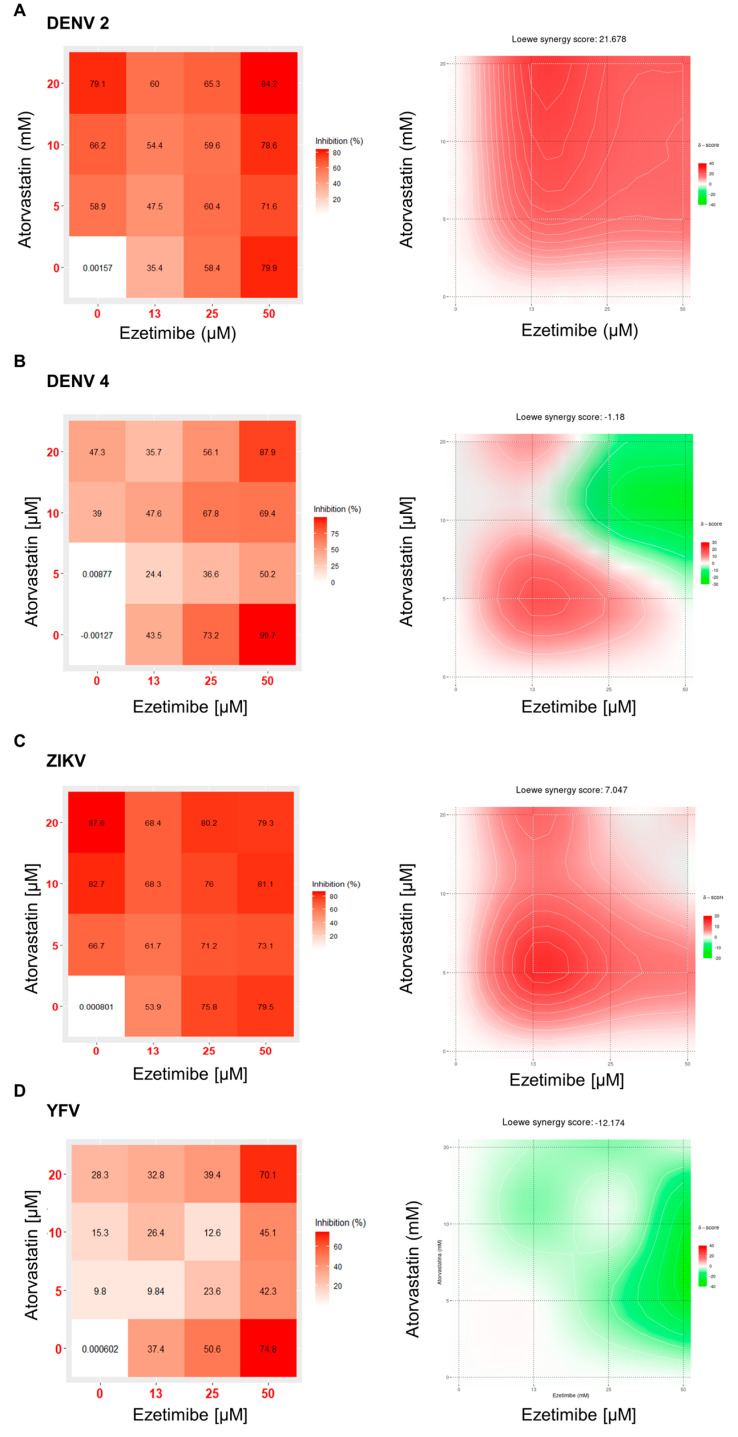
Analysis of the synergism effect between atorvastatin and ezetimibe against DENV-2 (**A**), DENV-4 (**B**), ZIKV (**C**), and YFV (**D**). The dose–response percentage inhibition matrices of the combined treatments of atorvastatin and ezetimibe and the interaction maps calculated using the Loewe additive model are shown. Areas with synergism scores of 10 or more (in red) represent synergy between the drugs.

**Figure 5 viruses-15-01465-f005:**
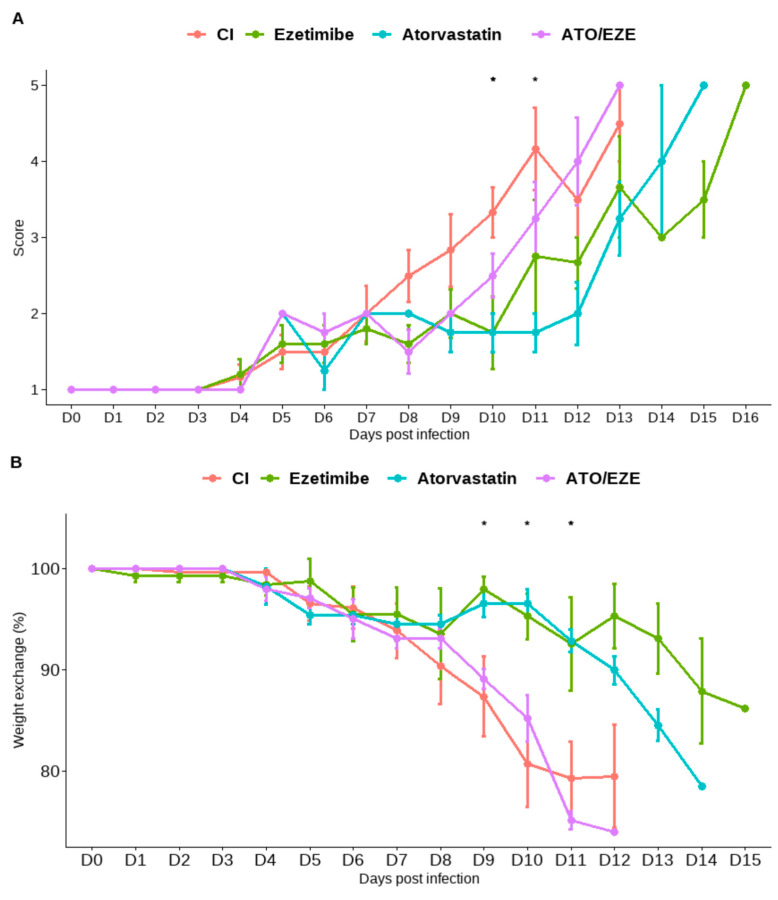
Effect of treatment with atorvastatin and ezetimibe in monotherapy and combined treatment in AG129 mice infected with DENV 2. Clinical sign scores (**A**) and percent change in weight (**B**) assessed daily from day of inoculation (D0) to euthanasia. Asterisks (*) shows statistical significance (* = *p* < 0.05).

**Figure 6 viruses-15-01465-f006:**
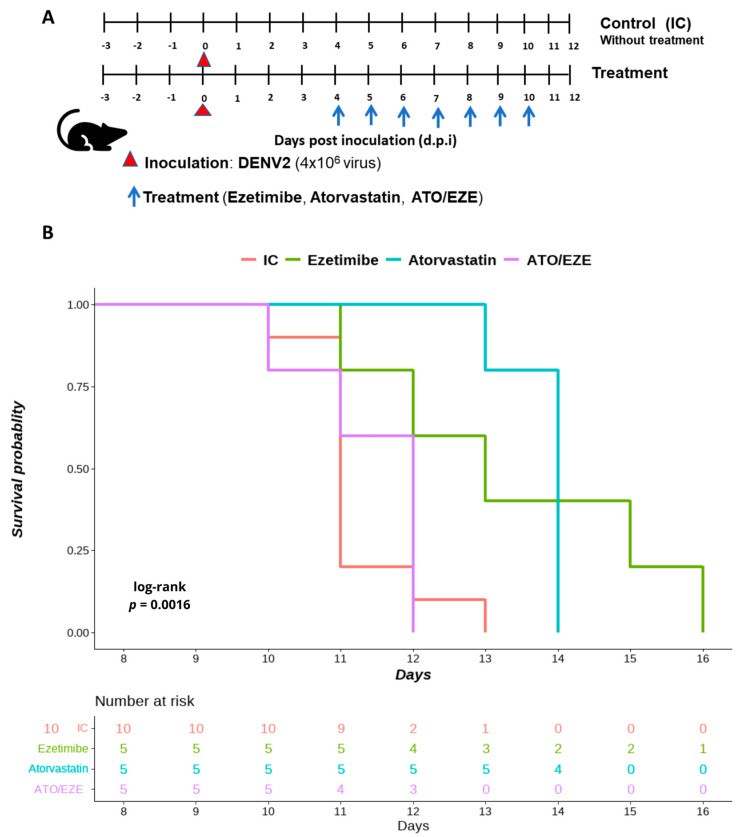
Survival analysis of DENV-2. infected AG129 mice treated with atorvastatin and ezetimibe. A schedule for in vivo experimental strategy is displayed (**A**). Survival probabilities for atorvastatin and ezetimibe are depicted using Kaplan–Meier curves (**B**).

**Table 1 viruses-15-01465-t001:** Selectivity Indexes (SI) for ezetimibe and atorvastatin against flaviviruses.

Virus	Ezetimibe	Atorvastatin
DENV 2	2.93 *	2.35
DENV 4	2.34 *	1.67
ZIKV	2.73	3.67
YFV	1.87	1.45

* Data extracted from Reference: [10].

## Data Availability

The datasets generated and analyzed during the study are available from the corresponding authors upon reasonable request.

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
