# Peer review of "Cholesterol-Lowering Drugs as Potential Antivirals: A Repurposing Approach against Flavivirus Infections"

_viruses, 2023, doi:10.3390/v15071465_

Round 1

Reviewer 1 Report

The authors submitted a manuscript titled: “Cholesterol-lowering Drugs as Potential Antivirals: A Repurposing Approach Against Flavivirus Infections ” for peer-review procedure in Viruses journal. The topic is of high interest since there is no comprehensive review addressing this topic available at present. There are some minor suggestions for the further improvement of this manuscript.

1.     Line 192-194. This sentence needs revision and correction. For example calculate should be replaced with calculated.

2.     L-233. There is typing mistake of “Lowet”. Please correct it

3.     In section 3.4 of results. Please explain the logic of using DENV-2-infected AG129 mice and why you have not used other flavivirus-infected AG129 mice.

4.     Line 264: A total of 25 animals were examined: 10 control mice without treatment, 5 mice treated with ezetimibe, 5 mice treated with atorvastatin, and 5 mice treated with both ezetimibe and atorvastatin. Why you have not divided mice equally in each group.

Author Response

The present manuscript is a revised version of the manuscript number viruses 2455772  (Cholesterol-lowering Drugs as Potential Antivirals: A Repurposing Approach Against Flavivirus Infections) which was submitted to Viruses. This revised version has been modified taking into consideration each of the valuable comments made by the reviewers. 

REVIEWER 1: 

The authors submitted a manuscript titled: “Cholesterol-lowering Drugs as Potential Antivirals: A Repurposing Approach Against Flavivirus Infections ” for peer-review procedure in Viruses journal. The topic is of high interest since there is no comprehensive review addressing this topic available at present. There are some minor suggestions for the further improvement of this manuscript. 

Reply: We are grateful for your review and feedback. Your valuable comments will be taken into account. Thank you for your time and recommendations. 

  1. Line 192-194. This sentence needs revision and correction. For example, calculate should be replaced with calculated. 

Reply: We appreciate your observation that this was corrected in the manuscript. 

  1. L-233. There is typing mistake of “Lowet”. Please correct it 

Reply: We appreciate your observation; this was corrected in the manuscript. 

  1. In section 3.4 of results. Please explain the logic of usingDENV-2-infected AG129 miceand why you have not used other flavivirus-infected AG129 mice. 

Reply:  

Thank you for your insightful comments and questions. We appreciate the opportunity to clarify these aspects of our study.  

The AG129 mice were kindly donated by Dr. Marco Antonio Meraz Ríos (Biomedicine Department, Center for Research and Advanced Studies (CINVESTAV-IPN-México), These mice are requested for several projects within the Center. Therefore, we prioritized evaluating both drugs’ anti-DENV effect, based on the synergistic effect of our in vitro results. 

Our study investigated the antiviral potential of cholesterol-lowering drugs, atorvastatin, and ezetimibe, in monotherapy and combination against DENV, ZIKV, and YFV. In vitro results demonstrated a significant dose-dependent reduction in the percentage of infected cells for both drugs. Interestingly, only DENV-2 showed a synergistic effect when both drugs were tested. This finding leads us to conclude that further in vivo studies should be conducted to evaluate the antiviral impact of atorvastatin and ezetimibe against DENV-2. Nevertheless, it is possible that similar techniques could be employed to investigate other flaviviruses. In addition, DENV 2 is one of the four dengue virus serotypes known to cause severe disease in humans. By concentrating on DENV-2, we intended to investigate therapies for one of the virus's most dangerous forms (1). 

4: A total of 25 animals were examined: 10 control mice without treatment, 5 mice treated with ezetimibe, 5 mice treated with atorvastatin, and 5 mice treated with both ezetimibe and atorvastatin. Why you have not divided mice equally in each group. 

Reply: Thank you for your insightful comments and for raising this important issue. We understand your concern about the unequal distribution of mice across the different groups in our study. 

The decision to use a larger control group was based on establishing a robust baseline for comparison with the treated groups. This is common in preliminary studies where the control group's data variability is unknown. By having a larger control group, we aimed to estimate the natural variability better and thus improve the reliability of our findings (2, 3). 

In addition, we would like to note that the availability of the AG129 mouse model was a limiting factor in our study. The AG129 mouse model has limited breeding stock, and each mouse requires supervision and PCR mutation corroboration. This significantly limited the number of animals we could use in our study. 

However, we acknowledge that this approach may raise questions about the balance of statistical power among the groups. To address this, we performed rigorous statistical analyses to ensure that our results are not biased due to the unequal group sizes. Specifically, multiple mean comparisons were conducted using the Wilcoxon test to compare each treatment group to the control in the in vivo model (2). This non-parametric test is robust to unequal sample sizes and does not assume a normal distribution, making it suitable for our study design. 

References: 

  1. Herbinger K., Froeschl G., Romano C., Cabidelle A., Junior C.. Serotype Influences On Dengue Severity: a Cross-sectional Study On 485 Confirmed Dengue Cases In Vitória, Brazil. BMC Infect Dis 2016;16(1).https://doi.org/10.1186/s12879-016-1668-y 
  1. Festing, M. F., & Altman, D. G. (2002). Guidelines for the design and statistical analysis of experiments using laboratory animals. ILAR journal, 43(4), 244-258. 
  1. Charan, J., & Kantharia, N. D. (2013). How to calculate sample size in animal studies?. Journal of pharmacology & pharmacotherapeutics, 4(4), 303.1 

Reviewer 2 Report

Report for the manuscript: Cholesterol-lowering Drugs as Potential Antivirals: A Repurposing Approach Against Flavivirus Infections by Juan Fidel Osuna-Ramos et al., 

The above mentioned manuscript reports the antiviral efficacy of Atorvastatin and Ezetimibe as monotherapy and in combination against different Flaviviruses. Overall, the study is extensive and sound; and the manuscript is well-written. I have a few comments which will help authors to further improve the quality of the manuscript. 

Figure 1 A/B and Figure 4 A appears to be redundant and generate similar information. Hence, authors can consider removing one. 

I have a comment for data reorganization. Authors can consider merging Atorvastatin and Ezetimibe in one graph with different color (Black/Gray) as shown in Figure 4 A.

Line  #50: Authors can consider the following reference for Drug repurposing approach (PMID: 33238464).

Line 51-52: antiviral properties against various viruses, including Flaviviruses. However, the citations are only for the Flaviviruses.  Authors should cite the references for the activity of ATO (PMID: 26915805) and EZE (PMID: 23266293) against other viruses. 

Author Response

REVIEWER 2: 

Report for the manuscript: Cholesterol-lowering Drugs as Potential Antivirals: A Repurposing Approach Against Flavivirus Infections by Juan Fidel Osuna-Ramos et al., 

The above-mentioned manuscript reports the antiviral efficacy of Atorvastatin and Ezetimibe as monotherapy and in combination against different Flaviviruses. Overall, the study is extensive and sound; and the manuscript is well-written. I have a few comments which will help authors to further improve the quality of the manuscript.  

  1. Figure 1 A/B and Figure 4 A appear to be redundant and generate similar information. Hence, authors can consider removing one.  

Reply: 

We greatly appreciate your comment, figure 4 was modified and one of the graphs was removed. The figure caption was also modified,  

  1. Authors can consider merging Atorvastatin and Ezetimibe in one graph with different color (Black/Gray) as shown in Figure 4 A. 

Reply: 

We appreciate your observation, and thank you for your comment. Figure 2 and 3 were merged into one graph (Figure 2) with a different color (Black/Grey) and color was added to the curves for better differentiation. 

  1. Line  #50: Authors can consider the following reference for Drug repurposing approach (PMID: 33238464). 
  1. Line 51-52: antiviral properties against various viruses, including Flaviviruses. However, the citations are only for the Flaviviruses.  Authors should cite the references for the activity of ATO (PMID: 26915805) and EZE (PMID: 23266293) against other viruses.  

Reply: Thanks for your comments. The suggested references were added between lines 50-52, and a brief description

Round 2

Reviewer 2 Report

Authors have made sufficient changes and the results are well-organized as suggested. I recommend that this manuscript should be accepted in its current form.